# Study of compositions of musks in different types secreted by forest musk deer (*Moschus berezovskii*)

**Tianxiang Zhang**, **Weijiang Jin**, **Shuang Yang, Yimeng Li, Meishan Zhang, Minghui Shi, Xiaobing Guo, Dawei Li, Baofeng Zhang, Shuqiang Liu** \*, **Defu Hu**\*

School of Ecology and Nature Conservation, Beijing Forestry University, Beijing, People's Republic of China

☯ These authors contributed equally to this work.
\* liushuqiang@bjfu.edu.cn (SL); hudf@bjfu.edu.cn (DH)

**Data Availability Statement:** All relevant data are within the paper and its Supporting Information files.

## Abstract

Musk is a secretion of the forest musk deer (*Moschus berezovskii*). Normal musk is a brown solid secretion with a light fragrance. In this study, abnormal types of musk, namely, white and black musks, were discovered during the musk collection process. Researchers have long been concerned with the components of musk. Herein, GC-MS, headspace solid-phase microextraction (HS-SPME), and nonmetric multidimensional scaling (NMDS) were used to analyze the nonpolar organic components, volatile organic components, and sample similarities among different musks, respectively. Abundant steroid hormones and proteins were also found in the musk. The steroid hormone concentrations were detected using a radioimmunoassay (RIA). Proteins in the samples were hydrolyzed and the amino acids concentrations were detected. The steroid hormone and amino acid concentrations in white musk were significantly lower than in normal and black musks (p<0.05). The components were subjected to NMDS analysis to understand the differences in components among different types of musk, with the results suggesting that white musk was different from normal and black musks.

## Introduction

In nature, many types of animals produce secretions through special glands, and many types of endocrine and exocrine secretion glands exist. Different glands have different physiological functions, with each gland having multiple functions. Thorough research has determined the complex functions of particular glands. An important function of exocrine secretions is to spread olfactory information [1]. Compared with wild animals, humans have various ways to express information, while scent-marking and identification are strengths of many animals. Many large mammals use anal gland secretions (AGS) to mark their scent and declare territory [2–5]. Male rodents have developed preputial glands, often using preputial gland secretions and urine to spread odors [6, 7]. Many researchers have explored various methods for studying the complicated components in animal secretions. The most commonly implemented analytical techniques are gas chromatography (GC) [8], gas chromatography–mass spectrometry

**Funding:** This work was supported by Beijing Municipal Nature Science Foundation (Grant No. 5202016) and National key R&D program of China (2018YFD0502204).

**Competing interests:** No authors have competing interests.

(GC-MS) [9–11], high-performance liquid chromatography (HPLC) [12], electrospray ionization MS (ESI-MS) [13], gel electrophoresis [13], thin-layer chromatography (TLC) [14], and headspace solid-phase microextraction–GC-MS (HS-SPME–GC-MS) [3, 5, 15].

In some species, secretions provide the females with effective evidence of health conditions. Male giant pandas (*Ailuropoda melanoleuca*) mark with their AGS frequently through the year. However, in breeding seasons, the components are more complex than in nonbreeding seasons. These different components can act as signals that allow females to select their partners [4, 16]. The blue petrel (*Halobaena caerulea*) can spread preen gland secretions on their feathers. The females will identify odors on the males' feathers to judge their health condition. Finally, they will choose healthy males to provide the next generation with strong immunity [17–19].

The forest musk deer (*Moschus berezovskii*) is a forest-inhabiting species native to southern and central China and northernmost Vietnam [20]. This species is characterized by solitude, territoriality, and high vigilance [21, 22]. Male individuals can produce a strong-smelling secretion to mark their territory and attract females [23, 24]. The musk gland of an adult male forest musk deer is located between its navel and genitals. This organ synthesizes, stores, and secretes musk. Musk is considered a pheromone for attracting females in the breeding season. Research has shown that sex hormones play important roles in the musk formation process [23]. The formation of musk is a long-time process which remain several months. The initial musk is liquid with a color in yellow [23, 24]. This process in musk sac is similar with cerumen in ear. Water content and components change in the formation process. Normally, the final musk of healthy adult males are a black–brown solid substance [20, 24]. However, among captive populations, some individuals secrete abnormal musk. Two types of abnormal musk have been identified, namely, white musk, which is white in color and has a sour and rotten odor, and black musk, which is a black semi-solid with a rotten smell. Captive forest musk deer have many health problems, such as parasitosis [25], diarrhea [22], and abscesses [26]. In this study, we have considered abnormal secretions as a type of disease and produced records of clinical cases. We have also studied the differences in components between normal and abnormal musk.

## Materials and methods

### Sample collection

This study is approved by the Institute Research Committee of School of Ecology and Nature Conservation, Beijing Forestry University.

### Sample collection

This study analyzed musk samples from adult male forest musk deer (*Moschus berezovskii*) maintained in Pien Tze Huang Forest Musk Deer Breeding Center (Fengxian, Shaanxi, China) for at least two years. The collection work started at middle of February and lasted for 14 days, when the components were considered stable at this month [23]. To collect musk samples, all tools were sterilized with alcohol in advance. Fresh musk samples were placed in a sterile centrifuge tube, labeled with the respective deer's ear tag number and collection time, and immediately placed in liquid nitrogen for transporting back to the laboratory. Scientific studies were conducted on samples within one week of collection.

The musk was categorized into three categories, as follows. Normal musk (NM) is the usual mature secretion of healthy musk deer, presenting a brown color and faint fragrance. Musk samples that did not show these characteristics were considered abnormal and classified as either white musk (WM) or black musk (BM). White musk is an abnormal musk with a white

color and acrid smell. Black musk is black in color, and has a muddy appearance and unpleasant smell.

## Analysis of Nonpolar Organic Components (NPOC)

The musk was dried using a dryer containing calcium sulfate. The drying process is at 60°C, no additional pressure for more than 3 hours, till the weight of musk no longer changes. A sample of the dried musk (50±2 mg) was dissolved in dichloromethane (1 mL), decanted into a disposable syringe, filtered through a micropore filter (Nylon, 25 mm, 0.22 μm; Jinteng, China), used for GC-MS analysis. A sample of the filtrate (2 mL) was placed in a sample bottle and inserted into the GC-MS system (Model QP2010; Shimadzu Corp., Kyoto, Japan) for quantitation. The operating conditions were as follows: Precolumn pressure, 60 kPa; splitless injection; inlet temperature, 250°C; interface temperature, 280°C; carrier gas, He; sample volume, 1 μL; ion source temperature, 250°C; gas flow rate, 1.0 mL min$^{-1}$; mass spectrometer chamber temperature, 230°C; quadrupole analyzer temperature, 150°C; electron impact energy, 70 eV; mass scanning range, 40.0–400.0 amu. The temperature program was as follows: 80°C for 1 min, increased to 245°C at 5°C min$^{-1}$, held at 245°C for 1 min, increased to 280°C at 10°C min$^{-1}$, and then held for 5 min [23, 27].

Post-analysis, the structures of fragmentation ions were compared with those in the mass spectral library of the National Institute of Standards and Technology. Chromatography retention times and other relevant data were also logged. Collectively, these data were used to verify the musk components.

## Analysis of volatile Organic Compounds (VOCs)

The musk sample (20±1 mg) was weighed precisely, placed in a 20-mL headspace bottle, and sealed using a sealing cover with a PTFE pad. The sample was then heated at 45°C for 2 min, which was then increased to 120°C at 5°C min$^{-1}$, then to 200°C at 10°C min$^{-1}$, and finally to 280°C at 15°C min$^{-1}$, holding for 10 min.

The mass spectrometry parameters were as follows: Inlet temperature, 250°C; interface temperature, 230°C; electron energy, 70 eV; solvent delay, 3 min; mass scanning range, 40.0–400.0 amu.

## Analysis of amino acid components

Two musk groups were prepared. The first was analyzed for its free amino acids, while the other group was subjected to hydrolysis before analysis of the total amino acids in the hydrolyzed musk. Samples of unhydrolyzed musk (50±2 mg) were accurately weighed and added to centrifuge tubes with phosphate-buffered saline (PBS) solution (800 μL, 0.02 M). To obtain hydrolyzed musk, the musk samples (50±2 mg) were accurately weighed, mixed with 0.1% HCl (800 μL), vortexed (IKA MS 3 digital, Sigma Aldrich, USA) for 5 min, and then centrifuged using an Eppendorf Centrifuge (3000 rpm; Rotana 460R, Hettich, Germany) for 10 min at 4°C. The supernatant was stored at −20°C until analysis.

Samples (400 μL) were added to a Hitachi Amino Acid Analyzer (Tokyo, Japan) equipped with a separation column containing ion-exchange resin #2622 PH (4L1426 Hitachi; 4.6 mm ID × 60 mm; 3-μm particles), operated at a column temperature of 57°C and flow rate of 0.35 mL min$^{-1}$. A ninhydrin reaction column (4.6 mm ID × 40 mm) was also used, operated at a reaction temperature of 135°C and flow rate of 0.3 mL min$^{-1}$. Samples were detected at 570 and 440 nm (pro) using a visible light spectrophotometer, with an injection volume of 20 μL and sample analysis time of 50 min.

## Analysis of steroid hormone levels

Musk were weighed at about 0.05 g, the weight of each sample was measured and recorded. Place the samples with two quartz beads in individual centrifuge tubes, followed by the addition of 90% (v/v) ethanol (5 mL). Set the tubes in a ball mill instrument (AM100S, Ants Scientific Instruments Co. Ltd., Beijing, China). The frequency of milling is 20 Hz. After 20 min, the tubes were centrifuged for 20 min at 2,500 rpm and the supernatants were collected.

The ethanol extracts were evaporated to dryness in a 60°C water bath, and PBS solution (1 mL, 0.02 M) was added to each tube, followed by agitation to recover the hormones. The tubes were then stored at −20°C.

Hormones levels were quantified by radioimmunoassay (RIA) using a GC-2016 performance counter (Anhui Zonkia Scientific Instruments Co. Ltd., Anhui, China). The parameters recommended by the cortisol reagent manufacturer (Beijing North Institute of Biotechnology, Beijing, China) were as follows: Sensitivity, $\leq 2.0$ ng mL$^{-1}$; intrabatch coefficient of variation, $< 10\%$; interbatch coefficient of variation, $< 15\%$.

## Statistical analysis

The components in the NPOC and VOC results were used as elements to analyze the differences and similarities of samples. The samples were arranged in a visual low-dimensional plane such that the distance between samples reflected the relationship between samples in the plane scatter plot to the greatest extent using nonmetric multidimensional scaling (NMDS) [28]. NMDS analysis based on the Bray–Curtis similarities was performed using the R program (version 3.5.2). Data significance analysis was performed using 'One-way ANOVA in SPSS v19.0 software (IBM, Armonk, NY, USA).

## Results

### Appearance of musks of different types

Musk samples were classified according to their general appearance (Fig 1). Normal musk (NM) was a black–brown solid secretion (Fig 1A), WM was an abnormal musk type with a white color and stable solid form (Fig 1B), and BM was an abnormal musk type with a black color and soft muddy consistency (Fig 1C).

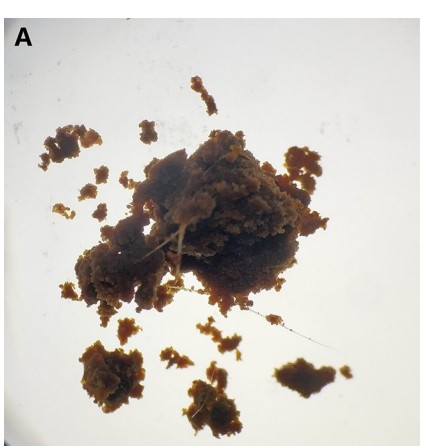 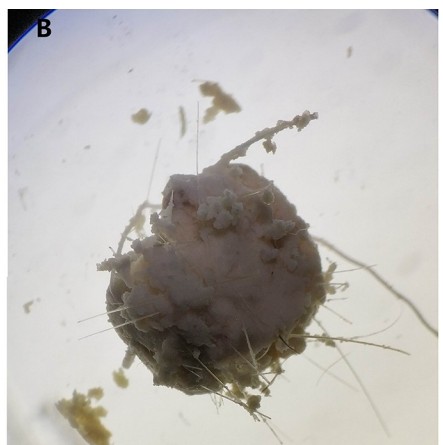 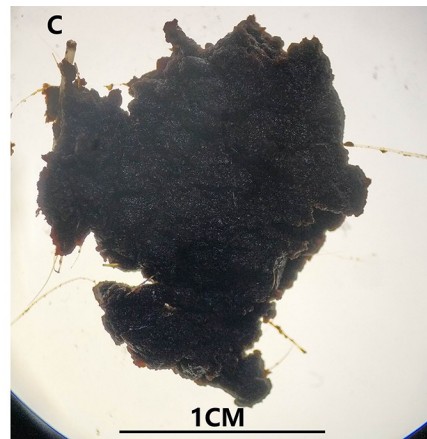

**Fig 1. Morphological comparison of different types of musk.** (**A**) Normal musk (NM); (**B**) white musk (WM); (**C**) black musk, (BM).

## Comparison of main components and cluster analysis

Among many components, 13 main components were selected. Compared with normal musk samples, significant differences in these 13 components were observed in the abnormal musk samples. The main components were muscone, heptadecanal, prasterone-3-sulfate, cholesta-3,5-diene, cholest-2-ene, lanosteryl tosylate, 4-methyl-3α-cholest-4-en-3-ol, cholesterol penta-fluoropropionate, cholestan-3-one, cholest-4-en-3-one, pentadecanal, docosyl pentafluoropro-pionate, and octadecyl trifluoroacetate. Compared with the NM group, muscone, prasterone-3-sulfate, lanosteryl tosylate, and pentadecanal showed significantly lower contents in the WM group (n = 5, $P<0.01$), while heptadecanal, cholesta-3,5-diene, cholest-2-ene, 4-methyl-3α-cholest-4-en-3-ol, cholesterol pentafluoropropionate, and docosyl pentafluoropropionate showed significantly higher contents in the WM group (n = 5, $P<0.01$).

## Comparison of VOCs

This box plot shows that the components of fatty acid types in WM is significantly more than those in NM and BM (n = 5, $P<0.05$). Fatty acids in WM group take a dominant position in all VOCs (average content >50%).

## NMDS analysis

In Fig 4, each point represents a sample, with different colors representing different types of musk. The distance between each two points represents the similarity of the two samples based on the Bray–Curtis algorithm. The ellipse represents the confidence interval for this dataset at a confidence level of 95%. In plot **A**, the BM group shared an intersection with most samples from the NM and WM groups. Samples from the NM group shared a small intersection with the WM group. In plot **B**, the BM group shared a large intersection with the NM group, and a small intersection with the WM group, while the NM group showed no intersection with the WM group.

## Analysis of steroid hormone levels

The cortisol concentrations of musk from the NM, BM, and WM groups were 3524.36 ±383.21, 2748.26±1404.17, and 843.55±523.45 ng $g^{-1}$(n = 5, Mean±se), respectively. The estra-diol concentrations of musk from the NM, BM, and WM groups were 115.00±3.96, 130.10 ±46.68, and 58.40±34.29 ng $g^{-1}$(n = 5, Mean±se), respectively. The testosterone concentrations of musk from the NM, BM, and WM groups were 182.71±86.95, 237.21±134.52, and 65.21 ±100.07 ng $g^{-1}$(n = 5, Mean±se), respectively. From these results, in general, the levels of the three hormones in the NM group were significantly different from those in the WM group ($P<0.01$). Meanwhile, the hormone levels in the BM group were significantly different from those in WM group, but not significantly different to those in the NM group ($P>0.05$).

## Differences in amino acid components

Fig 6 clearly shows that the free amino acid concentrations in musk were far lower than the total amino acid concentrations in hydrolyzed musk. The free amino acids concentrations in musk showed irregular tends. In contrast, in hydrolyzed musk samples, the concentrations of most amino acids in the NM and BM groups were significantly higher than those in the WM group (n = 5, $P<0.05$). Furthermore, most amino acid concentrations in the NM group were higher than those in the BM group.

## Discussion

Musk is a specific secretion only produced by male musk deer. Musk has strong and unique odors that helps the males mark their territory. Musk also plays an important role in the mating process as a pheromone that can demonstrate the charm of the males and attract females. In nature, many species use urine or secretions to mark territory. For example, Eurasian beavers (*Castor fiber*) use castoreum (dietary derivatives mixed with urine) and AGS to mark their territory [29, 30]. This behavior is the same between sexes, and partners usually cooperate, helping the beavers to strive for living space and avoid some unnecessary conflicts [31]. In some species, strong odors can not only help males to win territory, but also favor females. Female blue petrels can identify the odors on males' feathers produced by the preen gland secretions and judge whether they can provide a genome with strong immunity for the next generation [17–19].

As described above, the secretions provide a profile of male individuals in mating seasons. Females can select the ideal partner by judging and identifying the odors in the male secretions. However, in captivity, animals can be subjected to high stress [32]. Chronic stress can cause physical damage and decrease immunity. Therefore, captive animals are susceptible to physical and psychological problems [33, 34]. In captive forest musk deer populations, we observed that, during the period when musk is formed in the males' musk sac, it can be divided into different categories. Musk has previously been added to perfumes and cosmetics as an advanced fauna natural perfume [35–37]. However, abnormal musk has a bad smell that might make people feel uncomfortable and is, therefore, not qualified to be used as a perfume material (Fig 3). Musk extraction can be used in medicinal industry to obtain anti-inflammatory, antibacterial, anticoagulant, and other biological activities [38–40]. Muscone is regarded as the most important component in musk, and is responsible for the main bioactivity [41–43]. The components in normal musk and abnormal musk were different. For example, muscone was hard to find in white musk, but was the main component in normal musk (Fig 2). Furthermore, the contents of prasterone-3-sulfate, lanosteryl tosylate, and pentadecanal showed a positive correlation with muscone by comparison with the main components in the three groups. Meanwhile, the contents of heptadecanal, cholesta-3,5-diene, cholest-2-ene, 4-methyl-3α-cholest-4-en-3-ol, cholesterol pentafluoropropionate, and docosyl pentafluoropropionate showed a negative correlation with muscone. Former research has presented the comparations between musk of wild and domestic musk deer [44]. The results show that no significant differences could be observed in steroid concentration between wild and captive musk deer by PCA and cluster analysis. However, this research points out that $\delta^{13}C$ values in steroids between wild and captive musk have significant differences ($P$<0.01). Muscone was first extracted and reported in 1906 by Walbum [45]. The first artificial synthetic was achieved by Stoll in 1947 [46, 47]. Studying the components in natural musk might provide new ideas for the biosynthesis of muscone. In the WM group, muscone was hardly detected in the musk samples. However, the contents of some main components were significantly higher than those in the other two groups. The connections among these components requires further chemical study.

The NMDS analysis was a highlight of this study. NMDS analysis has been widely applied in ecology research, especially in biocenosis research. This method has been used to detect meaningful underlying dimensions and visualize similarities between the investigated components. The composition of musk is complex, especially the VOC components, which are the odor profiles for animal individual identification. Some other researchers have used the same method to analyze the similarities of the components in water [48], flower scent [49, 50], and soil [51]. The HS-SPME results were difficult to analyze, so the VOCs were classified and

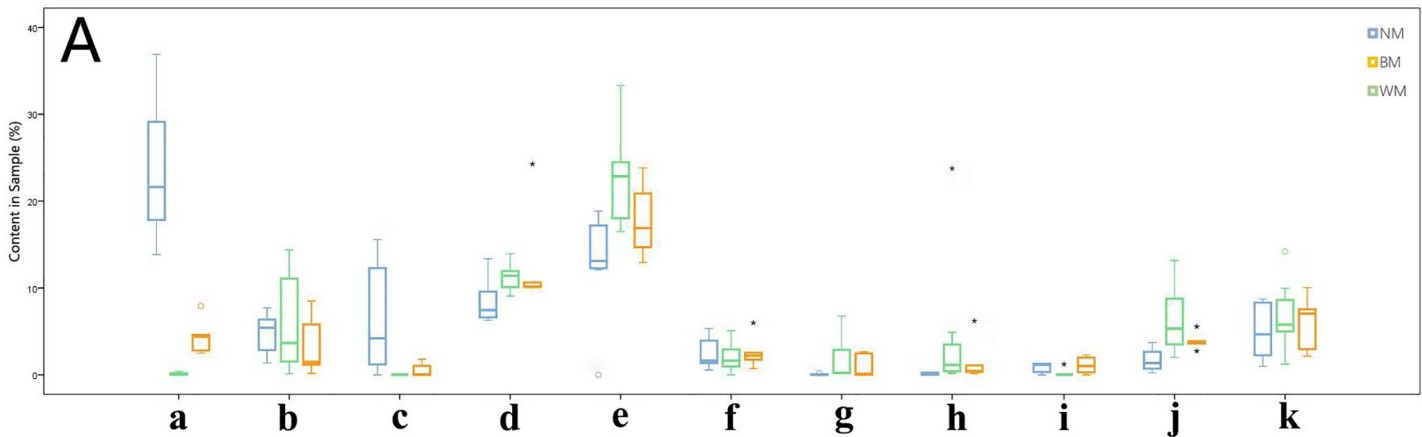

**Fig 2. Diagram of musk principal compositions.** (**A**) Box plot of principal compositions in different types of musk; (**B**) column diagram of musk compositions. * represents a statistically significant difference between the groups for each composition ($P < 0.05$). (a. muscone; b. heptadecanal; c. prasterone-3-sulfate; d. cholesta-3,5-diene; e. cholest-2-ene; f. lanosteryl tosylate; g. 4-methyl-3α-cholest-4-en-3-ol; h. cholesterol, pentafluoropropionate; i. pentadecanal; j. docosyl, pentafluoropropoonate; k. octadecyl trifluoroacetate).

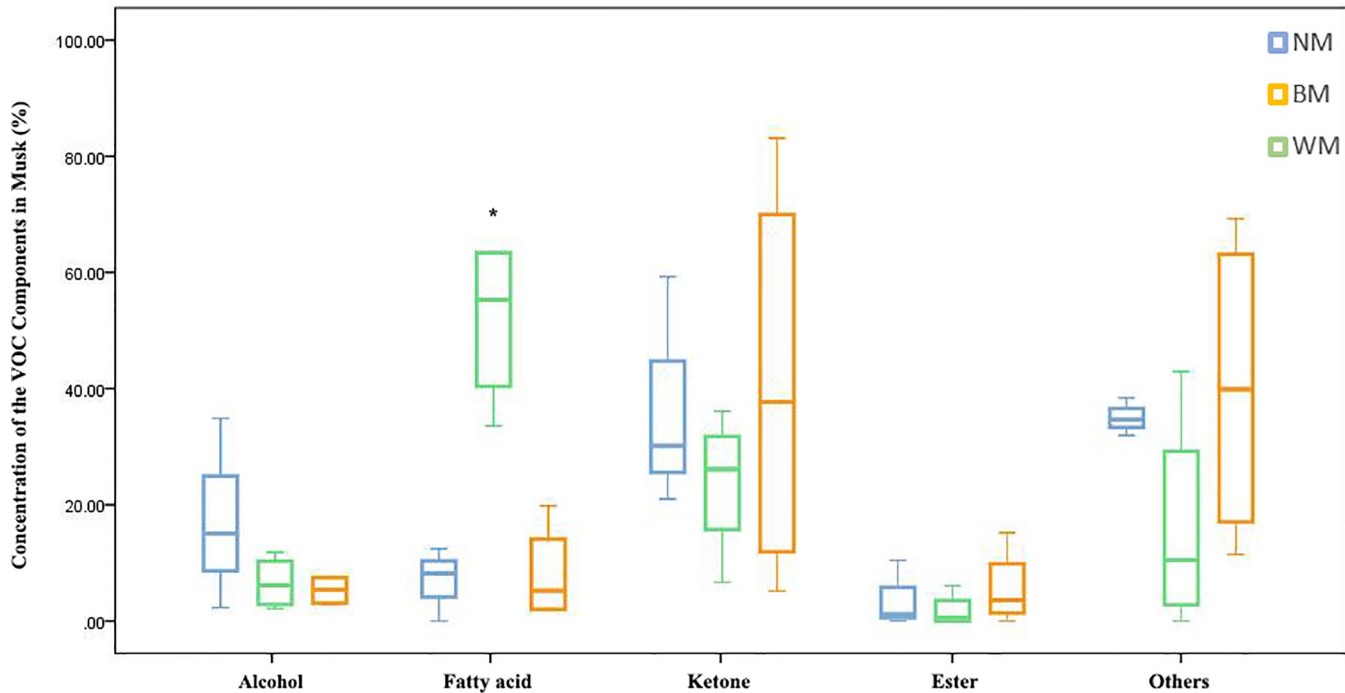

**Fig 3. Box plot of volatile components in different types of musk.** * Represents a statistically significant difference between the groups for each composition (*P*<0.05).

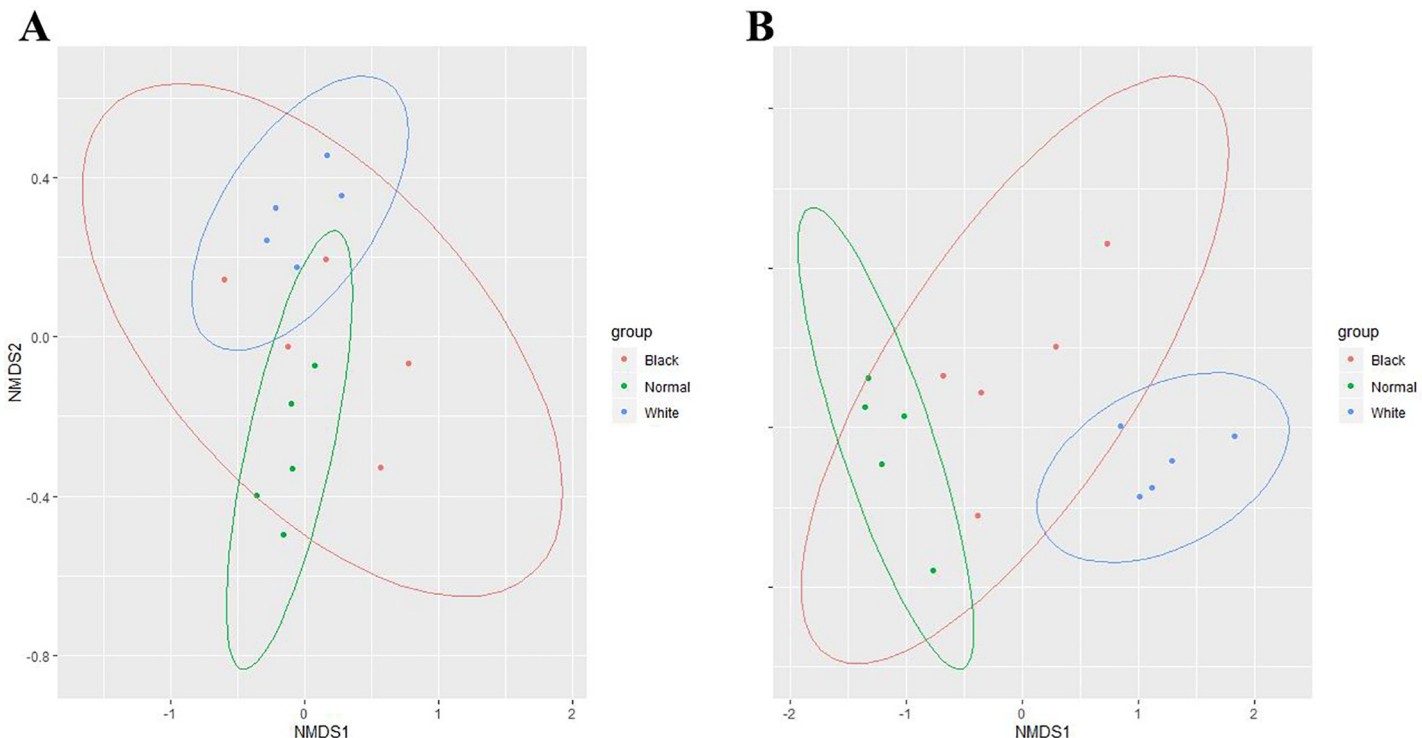

**Fig 4. NMDS plot of components in different musk samples.** (**A**) NMDS plot of NPOC; (**B**) NMDS plot of VOCs.

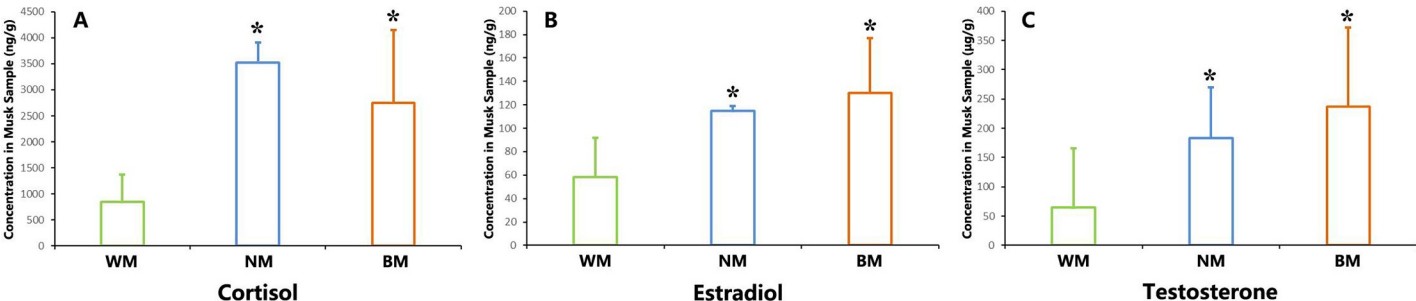

**Fig 5. Steroid hormone concentrations in musk ethanol extracts.** (**A**) Cortisol concentrations; (**B**) estradiol concentrations; (**C**) testosterone concentrations. *
Represents a statistically significant difference between groups for each composition (The column represents for the mean value of the group. Error bar represents for
the standard error of the group. _P_<0.05).

significant differences in the fatty acid concentration in the WM group were observed (Fig 3).
The component richness and high concentrations of fatty acids in WM might make the samples in the WM group have an unpleasant odor. The NMDS results showed a clear relationship
among the different types of musk (Fig 4). For NPOC (Fig 4A), samples from the BM group
showed similarities with both WM and NM groups. This suggested that the NPOC in black
musk varied considerably, and was an intermediate composition between white and normal
musks. Regarding the VOC (Fig 4B), the NM group was significantly different to the WM
group, while the BM group seemed to be more similar to the NM group. Furthermore, the
NMDS result help us to determine the similarity of the musks, beyond appearance alone.

Musk is a special secretion that is closely related to sexual behavior. Some researchers have
found that sex hormones play important roles in the musk formation process [23, 27, 52]. A
similar mechanism is found in peacock blenny (*Salaria pavo*), musk shrew (*Suncus murinus*),
muskrat (*Ondatra zibethicus*), and other species [53–56]. Sex hormones can regulate the function and components of the exocrine gland. We analyzed the sex hormone levels in musk, with
those in the WM group found to be significantly lower than those in BM and NM groups (Fig
5). Captive breeding is an efficient way to help the endangered species restore the population.
The potential stressors caused by the captive environment can cause health problems to the
animals [57]. Chronic stress is a common threaten to the health of captive animals, which can
cause physiological disorder [58]. This disorder can inhibit the secretion of cortisol in adrenal
cortex. Therefore, the cortisol level of the individuals suffering from chronic stress is significantly lower than the normal level. This physiological disorder brings the reduction in testosterone and estradiol levels. The cortisol level can be an important indicator to the health
condition of musk deer. The individuals which secrete normal musk may not be avoiding
from the threaten of chronic stress. But the individuals which secrete white musk can be confirmed suffering from chronic stress.

There are many studies on the components of exocrine secretions, because some components may provide good nutrition for microbes. To avoid bad infections, the gland will also
secrete antimicrobe peptides [59]. Secretions of muskrat contain amino acids, with the scented
glands concentrating amino acids during the secretion season [60]. In non-breeding seasons,
the scented glands of muskrat show low activity in amino acid metabolic pathways. Different
physical conditions can affect the exocrine gland and reflect the amino acids metabolism. As
another example, cerumen can be used as a noninvasive sample to determine health conditions
in pregnant women. Fourteen types of amino acid decreased significantly in the late pregnancy
and early lactation periods compared with the nonpregnant period [61]. In our study, the free
amino acids contents in musk before hydrolysis in the three groups seemed extremely low

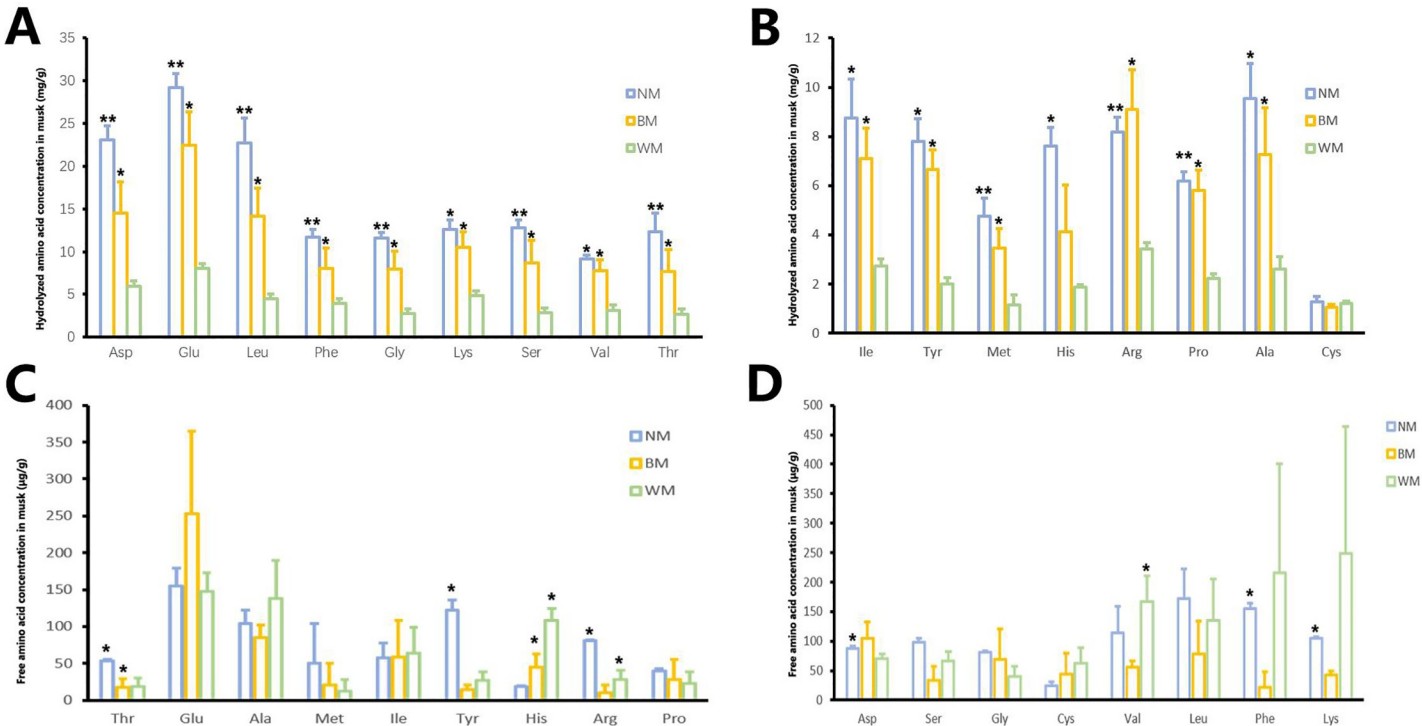

**Fig 6. Hydrolyzed amino acids and free amino acids detected in three types of musk.** (**A**, **B**) Amino acids detected in hydrolyzed musk; (**C**, **D**) free amino acids detected in musk. Bars of different colors represent different musk groups. **NM**, normal musk; **BM**, black musk; **WM**, white musk. Different symbols represent a statistically significant difference between the groups for each composition (The column represents for the mean value of the group. Error bar represents for the standard error of the group. $P$<0.05).

compared with the concentrations of total amino acids in hydrolyzed musk (Fig 6). Further-more, the amino acid concentrations showed a trend of enrichment in the NM group. This result suggested that significantly more proteins and peptides were present in normal musk than in abnormal musk. The amino acid contents in the normal musk showed a similar trend to amino acids in the muskrat musk [60]. The concentrations of Asp, Glu, and Leu were much higher in the total detected amino acids. In general, changes in the amino acid metabolism-related pathways in muskrat showed that amino acid metabolism provided the energy essential for musk secretion and breeding behavior. However, the functions of amino acid composi-tions, proteins, and peptides were not clear, perhaps owing to the microbes being expended for proliferation, or the forest musk deer itself not being strong to produce enough such components.

## Conclusions

We have discovered abnormal-type musk during the musk collection process and reported this phenomenon for the first time. Analysis of the total components of musk provided a greater understanding of musk. NMDS analysis helped to determine the similarity of different types of musk rather than judging by appearance. White musk was significantly different from normal musk and black musk, not only owing to the color and smell, but also according to component comparisons and similarity analysis, especially for the sex hormone levels in the musk. White musk can be an indicator to the health condition of musk deer. The individuals secrete white musk maybe suffering from chronic stress and more diseases.

## Supporting information

**S1 Data.**
(XLSX)

**S2 Data.**
(XLSX)

**S3 Data.**
(XLSX)

**S4 Data.**
(XLSX)

## Acknowledgments

The experimental facilities were provided by the School of Ecology and Nature Conservation Innovation Laboratory, Beijing Forestry University. We thank all the managers and breeders of the Pien Tze Huang Forest Musk Deer Breeding Center for supporting the sample collection. We thank Simon Partridge, PhD, from Liwen Bianji, Edanz Editing China (www. liwenbianji.cn/ac), for editing the English text of a draft of this manuscript.

## Author Contributions

**Data curation:** Weijiang Jin, Meishan Zhang, Xiaobing Guo.

**Formal analysis:** Weijiang Jin.

**Investigation:** Tianxiang Zhang, Minghui Shi, Dawei Li, Baofeng Zhang.

**Methodology:** Tianxiang Zhang, Weijiang Jin, Shuang Yang, Meishan Zhang, Dawei Li, Baofeng Zhang.

**Resources:** Tianxiang Zhang, Minghui Shi, Defu Hu.

**Software:** Yimeng Li, Minghui Shi.

**Supervision:** Shuqiang Liu, Defu Hu.

**Validation:** Shuqiang Liu, Defu Hu.

**Visualization:** Shuqiang Liu, Defu Hu.

**Writing – original draft:** Tianxiang Zhang.

**Writing – review & editing:** Tianxiang Zhang, Shuang Yang, Yimeng Li, Xiaobing Guo.

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
