## [Decision Letter · Decision Letter 0]

4 Dec 2020

PONE-D-20-31384

Study of Compositions of Musks in Different Types Secreted by Forest Musk Deer (Moschus berezovskii)

PLOS ONE

Dear Dr. Liu,

Thank you for submitting your manuscript to PLOS ONE. After careful consideration, we feel that it has merit but does not fully meet PLOS ONE’s publication criteria as it currently stands. Therefore, we invite you to submit a revised version of the manuscript that addresses the points raised during the review process.

We look forward to receiving your revised manuscript.

Kind regards,

Joseph Banoub, Ph,D., D. Sc., FRSC

Academic Editor

PLOS ONE

Journal Requirements:

2.We suggest you thoroughly copyedit your manuscript for language usage, spelling, and grammar. If you do not know anyone who can help you do this, you may wish to consider employing a professional scientific editing service.  

Reviewers' comments:

Reviewer's Responses to Questions

**Comments to the Author**

1. Is the manuscript technically sound, and do the data support the conclusions?

Reviewer #1: Yes

Reviewer #2: Partly

Reviewer #3: Partly

2. Has the statistical analysis been performed appropriately and rigorously? 

Reviewer #1: Yes

Reviewer #2: Yes

Reviewer #3: No

3. Have the authors made all data underlying the findings in their manuscript fully available?

Reviewer #1: Yes

Reviewer #2: Yes

Reviewer #3: Yes

4. Is the manuscript presented in an intelligible fashion and written in standard English?

Reviewer #1: Yes

Reviewer #2: Yes

Reviewer #3: Yes

5. Review Comments to the Author

Reviewer #1: Normal musk is a brown solid secretion with a light fragrance and it is secreted by the forest musk deer (Moschus berezovskii)...

In this manuscript, the authors report the discovery of two abnormal types of musk, namely the white and black musks, which were never been reported in the literature. The musk’s secreted during the breeding season act as pheromone signals which serve to inform the females about their state of health and their reproductive capacities.

The authors have used GC-MS, headspace solid-phase microextraction (HS-SPME), and nonmetric multidimensional scaling (NMDS) were used to analyze and quantify the nonpolar organic components, volatile organic components, and sample similarities among different musks.

The authors have also detected and identified the abundant steroid hormones using a radioimmunoassay (RIA) Furthermore, the musk proteins were hydrolyzed and the amino acids concentrations and detected.

The authors found that the steroid hormone and amino acid concentrations in white musk were significantly lower than in normal and black musk’s (p<0.05).

The authors concluded that the NMDS analysis permitted to understand the differences in components among different types of musk.

To sum it up, this research complements several previous works on these same animal (references 37-41). The chemical composition of the three musks observed and their consequences on the biological level are well analyzed. This work can be accepted as it is

Reviewer #2: The study presents original research in providing GC-MS, HS-SPME, NMDS and amino acid analysis data to characterise and distinguish between ‘normal’ and ‘abnormal’ musks, which the authors defined in the manuscript. The statistical analysis of the data seem appropriate for the study. There are some points in the experimental that should be clarified, e.g., In line 1, under the subheading ‘Analysis of nonpolar organic components’ the drying condition should be made in more detail e.g. was drying completed to constant mass, also what was the temperature, time and pressure (e.g. vacuum ?) used? The term ‘weighed precisely’ should be made clearer e.g. ± mg. Under the heading ‘Analysis of steroid levels’ mentions ‘milling’; therefore details should be given, such as the type of mill and time over which milling was performed.

The paper presents an extensive and comprehensive coverage of the literature more in the frame of a review but in the discussion, there is a lack of focus on explaining the significance of differences between the musks studied and speculations shown be offered into the significance in the finds of the abnormal musk relative to normal musk. The conclusion should give more in emphasizing the key differences that the authors have found and summarizing the major aspects of the ‘understanding’ they have made and indicate what is the impact for the reader, e,g. pointing to key distinguishing identifiers and such as possible importance to future deer stock.

The article is presented in an intelligible fashion and is written in standard English.

It is recommended that the paper by published subject to the points in my review being addressed.

Reviewer #3: In the manuscript titled “Study of Compositions of Musks in Different Types Secreted by Forest Musk Deer (Moschus berezovskii)”, Zhang et al. examined the composition of musk collected from M. berezovskii deer that were kept in a breeding center. Specific comments are as follows:

1) The first couple of paragraphs of the Introduction do not provide a reader insight into what the study is about. Perhaps shorten and move the content of the first couple of paragraphs, and make sure more emphasis is put on musk.

2) As musk composition will differ depending on time of year of collection, when were samples collected? Provide that information.

3) What was the health status of the deer from which white and black musk was obtained? Did deer with normal musk have any health concerns?

4) As deer were in captivity, are there data from non-captive deer for comparison? If not, provide a discussion on this issue.

5) For all data shown, provide the n value and indicate if the bars represent SE or SD. In addition, given that p values are being reported to be only less than 0.05, it raises concern that not all of the approx. 1000 samples were examined. If this is the case, why?

6) For all data shown with three groups, an ANOVA should be used with a comparison of each group to each other, and not a t-test. Please revise.

7) This reviewer is unclear about the X-axis labeling for Figure 2B. Please clarify.

6. PLOS authors have the option to publish the peer review history of their article (what does this mean?). If published, this will include your full peer review and any attached files.

Reviewer #1: No

Reviewer #2: No

Reviewer #3: No

---

## [Author Response · Author response to Decision Letter 0]

5 Jan 2021

Reviewer #1: Normal musk is a brown solid secretion with a light fragrance and it is secreted by the forest musk deer (Moschus berezovskii).

In this manuscript, the authors report the discovery of two abnormal types of musk, namely the white and black musks, which were never been reported in the literature. The musk’s secreted during the breeding season act as pheromone signals which serve to inform the females about their state of health and their reproductive capacities.

The authors have used GC-MS, headspace solid-phase microextraction (HS-SPME), and nonmetric multidimensional scaling (NMDS) were used to analyze and quantify the nonpolar organic components, volatile organic components, and sample similarities among different musks.

The authors have also detected and identified the abundant steroid hormones using a radioimmunoassay (RIA) Furthermore, the musk proteins were hydrolyzed and the amino acids concentrations and detected.

The authors found that the steroid hormone and amino acid concentrations in white musk were significantly lower than in normal and black musk’s (p<0.05).

The authors concluded that the NMDS analysis permitted to understand the differences in components among different types of musk.

To sum it up, this research complements several previous works on these same animal (references 37-41). The chemical composition of the three musks observed and their consequences on the biological level are well analyzed. This work can be accepted as it is

Response: Thanks for your kind comments, we have revised the manuscript and we are glad to see your further comments.

 

Reviewer #2: The study presents original research in providing GC-MS, HS-SPME, NMDS and amino acid analysis data to characterise and distinguish between ‘normal’ and ‘abnormal’ musks, which the authors defined in the manuscript. The statistical analysis of the data seem appropriate for the study. There are some points in the experimental that should be clarified, e.g., In line 1, under the subheading ‘Analysis of nonpolar organic components’ the drying condition should be made in more detail e.g. was drying completed to constant mass, also what was the temperature, time and pressure (e.g. vacuum ?) used? The term ‘weighed precisely’ should be made clearer e.g. ± mg. Under the heading ‘Analysis of steroid levels’ mentions ‘milling’; therefore details should be given, such as the type of mill and time over which milling was performed.

Response: We added some descriptions about the drying condition as: ‘The drying process is at 60 °C, no additional pressure for more than 3 hours, till the weight of musk no longer changes.’

We added the information about ‘weighed precisely’, e.g. 50±2 mg.

We added some descriptions about milling process as: ‘Set the tubes in a ball mill instrument (AM100S, Ants Scientific Instruments Co. Ltd., Beijing, China). The frequency of milling is 20 Hz.

’

The paper presents an extensive and comprehensive coverage of the literature more in the frame of a review but in the discussion, there is a lack of focus on explaining the significance of differences between the musks studied and speculations shown be offered into the significance in the finds of the abnormal musk relative to normal musk. The conclusion should give more in emphasizing the key differences that the authors have found and summarizing the major aspects of the ‘understanding’ they have made and indicate what is the impact for the reader, e,g. pointing to key distinguishing identifiers and such as possible importance to future deer stock.

Response: Thanks for your suggestions. We added some discussion on the chronic stress problem and relationships between chronic stress and cortisol levels. We mention that the musk deer secrete white musk may suffer from chronic stress problem in the CONCLUSION section.

The article is presented in an intelligible fashion and is written in standard English.

Response: Thanks for your kind comments. We revised the manuscript. We are glad to see your further comments.

It is recommended that the paper by published subject to the points in my review being addressed.

 

Reviewer #3: In the manuscript titled “Study of Compositions of Musks in Different Types Secreted by Forest Musk Deer (Moschus berezovskii)”, Zhang et al. examined the composition of musk collected from M. berezovskii deer that were kept in a breeding center. Specific comments are as follows:

1) The first couple of paragraphs of the Introduction do not provide a reader insight into what the study is about. Perhaps shorten and move the content of the first couple of paragraphs, and make sure more emphasis is put on musk.

Response: We delete some description in the Introduction section. We add some description about the appearance and formation process about musk in the Introduction section.

2) As musk composition will differ depending on time of year of collection, when were samples collected? Provide that information.

Response: Thanks for your suggestion. We add such information in the Sample collection section. This work began at the middle of the February, when the components of musk were considered stable. This work costed about two weeks.

3) What was the health status of the deer from which white and black musk was obtained? Did deer with normal musk have any health concerns?

Response: Most of the normal and black musk providers seemed no significant health problems. They were quite healthy at that time. Some of those seemed to be energetic. However, some white musk providers seemed to be weak at that time. Some of the white musk providers had diarrhea or hair slip problems. The other white musk providers seemed no significant problems but weak.

4) As deer were in captivity, are there data from non-captive deer for comparison? If not, provide a discussion on this issue.

Response: We didn’t have such sample from wild species. We found other researcher have studied the components differences between musk from domestic and wild musk deer. We cited this reference in the Discussion section and have some discussion on this study.

5) For all data shown, provide the n value and indicate if the bars represent SE or SD. In addition, given that p values are being reported to be only less than 0.05, it raises concern that not all of the approx. 1000 samples were examined. If this is the case, why?

Response: The n value of all analysis is 5. We chose 5 stable samples for all analysis. We mentioned that we collected over 1000 samples. In order to let the readers to understand that abnormal musk was infrequent samples in the whole community (20 white musk and 24 black musk). But we only took 8 normal musk back to the laboratory. Including 20 white musk samples and 24 black musk samples, most of them could not meet the require of amount in this study. At last, we chose 5 samples from each group to complete the whole study.

6) For all data shown with three groups, an ANOVA should be used with a comparison of each group to each other, and not a t-test. Please revise.

Response: Thanks for your correction. It was a misunderstand between our co-authors. We actually use one-way ANOVA to analysis the difference between two groups.

7) This reviewer is unclear about the X-axis labeling for Figure 2B. Please clarify.

Response: Did you mean the labels under X-axis of Figure 2A? We added some information in the figure legends of Figure 2.

---

## [Editor Report · Decision Letter 1]

6 Jan 2021

Study of Compositions of Musks in Different Types Secreted by Forest Musk Deer (Moschus berezovskii)

PONE-D-20-31384R1

Dear Dr. Liu,

We’re pleased to inform you that your manuscript has been judged scientifically suitable for publication and will be formally accepted for publication once it meets all outstanding technical requirements.

Kind regards,

Joseph Banoub, Ph,D., D. Sc., FRSC

Academic Editor

PLOS ONE

Additional Editor Comments (optional):

The authors have answered all the queries demanded by the referees
---

## [Editor Report · Acceptance letter]

29 Jan 2021

PONE-D-20-31384R1 

Study of Compositions of Musks in Different Types Secreted by Forest Musk Deer (*Moschus berezovskii*) 

Dear Dr. Liu:

I'm pleased to inform you that your manuscript has been deemed suitable for publication in PLOS ONE. Congratulations! Your manuscript is now with our production department. 

Kind regards, 

on behalf of

Dr. Joseph Banoub 

Academic Editor

PLOS ONE